# Spatial Distribution and Sources of Organic Matter in the Surface Sediments of Fuxian Lake, SW China

**Kai Zhang [1,2], Dongli Li [3], Xuejun He [2], Changyuan Xie [4] and Haibo He [2,4,*]**

1   School of Mechanical and Electronic Engineering, Hefei Technology College, Hefei 238000, China
2   State Key Laboratory of Environmental Geochemistry, Institute of Geochemistry, Chinese Academy of Sciences (CAS), Guiyang 550081, China
3   Institute of Surface-Earth System Science, School of Earth System Science, Tianjin University, Tianjin 300072, China
4   Yunnan Key Laboratory of Earth System Science, Yunnan University, Kunming 650500, China
*   Correspondence: hehaibo@mail.gyig.ac.cn

**Abstract:** Sedimentary organic matter is an important component of the metabolism of a lake's ecosystem, and it is generally derived from both the watershed and the primary productivity of a lake. Understanding the sources of organic matter in lakes and lake trophic status is important when evaluating the quality of lake ecosystems. We summarize the spatial distribution of total nitrogen (TN), total organic carbon (TOC), TOC/TN (C/N) molar ratios, and organic carbon isotope ($\delta^{13}C_{org}$) of the surface sediments of Fuxian Lake, Yunnan–Guizhou Plateau, Southwest China, which is the second deepest freshwater oligotrophic lake in China. The results show that the distributions of TN, TOC, C/N, and $\delta^{13}C_{org}$ of the surface sediments are spatially heterogeneous, which is also the case for the trophic conditions of the lake. Compared with the adjacent eutrophic lakes and typical lakes in other areas with strong human activities, the content of organic matter is at a low level. Meanwhile, the autochthonous organic carbon in the surface sediments was characterized by lower $\delta^{13}C_{org}$ ($-25.3 \sim -28.5$) and C/N ($8.7 \sim 12.9$), suggesting that the biological carbon pump effect plays a significant part in the stability of carbon sinks by coupling with carbonate weathering. Our results emphasize the importance of the carbon sink of coupled carbonate weathering and aquatic photosynthesis in the evolution of the carbon cycle in lakes. Although modern monitoring shows that Fuxian Lake is an oligotrophic lake, there are potential risks of organic nitrogen pollution with respect to surface sediments, especially in northern and southern shallow-water areas. The organic pollution of lakes can be reduced by controlling the discharge of wastewater and reducing the nutrient loading of agricultural runoff.

**Keywords:** lake ecosystems; surface sediments; organic matter sources; carbon isotopes; distribution characteristics

## 1. Introduction

With global climate change, the greenhouse effect has led to an increase in $CO_2$ in the atmosphere and caused a large increase in primary production in inland lakes [1]. Although many studies believe that freshwater systems are $CO_2$ sources [2], this is not always the case (including but not limited to): (1) While the dynamic equilibrium of a water body $CO_2$ depends on the physical process, the diffusion rate is only one in ten thousand of that in the atmosphere [3]; (2) A unique physiochemical characteristic of karst water is that its $CO_2$ concentration falls below 1% of dissolved inorganic carbon (DIC) when the pH level exceeds 8 [4]; (3) There are large amounts of recalcitrant dissolved organic carbon in karst aquatic ecosystems (decreased release of carbon compounds) [5]; (4) Carbon cannot meet the growing demands of highly eutrophic lakes [6]; and (5) $CO_2$ can be seasonally and locally depleted when photosynthesis occurs [4]. Therefore, the problem of carbon sinks in terrestrial aquatic ecosystems deserves further attention.

Researchers have found that atmospheric $CO_2$ can be converted into organic carbon via a series of physical, chemical, and biological processes and can be buried in sediments [7–10]. Compared to allochthonous sources [11], the organic matter (OM) component produced in situ by aquatic plants is termed autochthonous [9]; this OM is synthesized from dissolved $CO_2$ ($CO_{2aq}$), either photosynthetically or chemosynthetically, by submerged macrophytes, algae, and other microbial organisms. If the resulting autochthonous OM is deposited and buried in lake sediments, it can remove carbon from the short-term carbon cycle and form long-term stable carbon storage, which is similar to the marine biological carbon pump (BCP) effect [12]. Recent studies have revealed a BCP effect in which aquatic photosynthesis takes up DIC derived from carbonate weathering, and part of the generated autochthonous organic carbon is buried [5,9,10,13–16]. Researchers have presented the carbon sink mechanism of the carbonate-weathering-derived carbon sink coupled with aquatic photosynthesis (coupled carbonate weathering (CCW)), which indicates that carbonate weathering can form long-term effective carbon sinks via organic carbon burial [14,17].

Lakes are a significant component of terrestrial ecosystems, and their sediments provide a rich archive of information on the environmental history of lake watersheds [7,8]. The organic matter (OM) component of lake sediments is widely used to reconstruct paleoclimates and paleoenvironments, and to evaluate the anthropogenic impacts on local and regional ecosystems [9]. The transport of OM to lakes is a significant link in the global carbon cycle [10]. In addition to the significant part of the OM of lake sediments in terrestrial aquatic ecosystems, its significance is also reflected in its participation in nutrient cycles and the regulation of lake ecosystems [18]. As shown in Figure 1, contemporary anthropogenic activities such as land-cover changes and agricultural land-use have resulted in increased carbon (C) and nitrogen (N) discharge from soils into freshwater ecosystems [19–22], and thus strongly influenced the patterns of organic carbon (OC) input and trophic status [15,16]. Therefore, nutrient metabolism and the associated biogeochemical cycles are tightly coupled with aquatic photosynthetic production. It is well-known that in the transformation and flux of C and N, the main carrier is sedimentary organic matter [7]. In lake aquatic ecosystems, there is a coupled relationship between the water column and biogeochemical cycling in surface sediments [23]. Nutrients recycled within the sediments are liberated and subsequently diffused, and can then be supplied back to the overlying water and affect primary production via a somewhat complicated network of microbially-mediated biochemical reactions (Figure 1) [24].

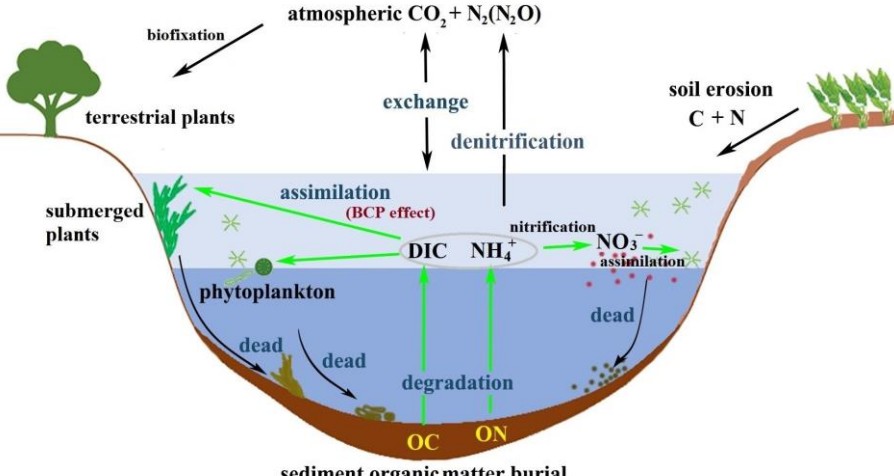

**Figure 1.** Schematic diagram showing the biogeochemical cycles influencing the organic matter of lake sediments.

In sum, a comprehensive understanding of the source and migration of OM in lacustrine sediments can help promote our knowledge of the global C cycle and clarify the coupled relationship between the C–N biogeochemical cycle and the related ecological and

environmental effects. This understanding may then provide technological and scientific support for the management and service of eutrophic lakes.

## 2. Significance of the Spatial Distribution of Organic Matter Sources in Lake Sediments

Surface lake sediments have long been considered as the main reservoir of OM, buried in terrestrial aquatic ecosystems. Lakes, especially carbonate lakes, are an important carbon sink in terrestrial ecosystems [8]. The large-scale buried OM in sediments is largely accomplished by the coupling of high sediment deposition/accumulation rates [8] and high storage efficiency, which is 50 times greater than the ocean on average [25]. For lakes, especially large lakes, with complex sedimentary environments, it is important to recognize the spatial distribution of sediments and their composition when assessing a lake's biogeochemical cycles; this is because the concentration and source may be spatially heterogeneous [26]. In addition, large quantities of nutrients may be released from fine-grained organic sediments at specific locations within the water body [27]. Therefore, it is necessary to map the distribution of sediments and their nutrient composition to create a spatially continuous surface in order to determine specific locations that will be targeted in the restoration of eutrophic lakes.

Various regional studies have confirmed the spatial heterogeneity of the organic components of lake sediments (e.g., in Australia (Coombabah Lake [23]), Asia (Nam Co Lake [28]), Europe (Sarbsko Lake [29]), and America (Pyramid Lake [30]). Numerous factors may influence the quantity of OM including the influx of terrestrial OM to the lake, aquatic productivity, various sediment properties, redox conditions, and the rate of microbial activity [23]. Generally, contributions from allochthonous and autochthonous OM, namely, from aquatic and terrestrial sources, have a direct–indirect impact on the spatio-temporal distribution of OM [26,29]. Thus, differentiating autochthonous and allochthonous sources of OM is a significant first process when evaluating the OM content of lake sediments.

## 3. Application of Tracer Methods in Determining the Sources of OM in Lake Surface Sediments

The diversity of OM sources depends not only on the biogeochemical characteristics of the lake watershed, but also on the social development status of urbanization and industrialization. Due to the large number of sinks and sources of OM in lake sediments, as above-mentioned, determining the constraints on its distribution is a challenging task [31]. However, as the major active component, OM is ultimately derived from allochthonous sources (terrestrial plant debris) or autochthonous sources (in situ primary production) in aquatic systems. Multiple bulk sediment parameters and molecular biomarkers have long been used to clarify the source of OM including C/N (atomic) ratios, stable organic carbon isotope ratios of OC ($\delta^{13}C_{org}$) [32], molecular biomarkers [33], and the combined use of stable carbon isotopes ($\delta^{13}C$) and radiocarbon isotopes ($\Delta^{14}C$) in sediments [34].

Any tracer method has its specific attributes including flaws. Essentially, the utilization of various parameters as source indices relies upon different OM sources having markedly different signatures. Conventional geochemical methods (e.g., $\delta^{13}C_{org}$ and C/N) have often been used to clarify the sinks and sources of OM within lacustrine environments [35]. The C/N molar ratios of bacteria, terrestrial plants, soil, phytoplankton, and aquatic macrophytes fall within a specific range [36]. In general, phytoplankton contain more proteins but are low in fiber, and the molar ratios are usually less than 10 [35]; the protein contents of emergent macrophytes and terrestrial plants are generally low, but the fiber contents are rich, which leads to a relatively high C/N molar ratios (>20) [35]. The study found that the C/N molar ratios of phytoplankton vary from 5 to 8, with an average of ~6 in submerged plants of 12 or more [36]. The $\delta^{13}C_{org}$ values for $C_4$ plants fall between −17‰ and −9‰, while that of terrestrial $C_3$ plants is relatively negative, between −32‰ and −20‰ [35]. Macrophytes have more positive $\delta^{13}C_{org}$ values (−30‰ to −5‰) due to their $C_4$-like photosynthesis pathway [37], while phytoplankton have relatively low $\delta^{13}C_{org}$ values (−42‰ to −20‰) due to $C_3$-like photosynthesis, which leads to a large degree of

$\delta^{13}$C fractionation [38]. Microbial sources (e.g., bacteria) in lake sediments can be indicated by their low $\delta^{13}C_{org}$ values ($-26\%$) [39]. However, the $\delta^{13}C_{org}$ values are influenced by various environmental factors that may lead to ambiguous and overlapping signatures, and C/N is directly affected by the mineralization of OM. Nevertheless, Wilson et al. (2005) [40] considered that the combined use of C/N molar ratios and $\delta^{13}C_{org}$ was an effective means of identifying OM sources. In general, the combined use of multiple indices can be expected to increase the degree of accuracy when determining the source of OM.

We compiled research on the sources and composition of OM in the surface sediments of Fuxian Lake, the largest lake, and a karst rift lake at the source area of the Pearl River. Although Fuxian Lake is oligotrophic, several studies have shown that it is at risk of becoming polluted [41]. Several previous studies have been conducted on the source, migration, and transformation processes and mechanisms of OM in Fuxian Lake [20,42] as well as on the stability of a carbonate weathering carbon sink [6,9,13]. In order to clarify the impact of anthropogenic activities and regional environmental parameters on the ecosystem of Fuxian Lake, we analyzed the spatial distribution characteristics of TN, TOC, $\delta^{13}C_{org}$, and C/N in the surface sediments of Fuxian Lake, and we used the results to distinguish the main sources of OM at different water depths to provide a basis for an environmental assessment of the lake.

## 4. Spatial Distribution of TN, TOC, $\delta^{13}C_{org}$, and C/N of the Surface Sediments and Their Significance for Organic Carbon Sources

Fuxian Lake is a freshwater lake that is located within the Yunnan–Guizhou Plateau ($24°21'$–$24°37'$ N, $102°49'$–$102°57'$ E, Southwest China). It is large (212 km$^2$) and deep (159 m) but has a relatively small catchment (675 km$^2$); the lake water is oligotrophic. The basin is shallow in the south and deep and wide in the north; there are considerable amounts of spring-fed inflows and small streams; the Liangwang River is the major inflowing stream, and Haikou River is the only outlet (Figure 2). The estimated retention time of Fuxian Lake is 167 years [43]. The present-day climate in this catchment is mainly controlled by the Indian Summer Monsoon (ISM). The average annual precipitation and temperature of this region are 951 mm and 5.6 °C, respectively, and more than 80% occurs in May and October [5].

The hydrochemistry of Fuxian Lake varies seasonally, as shown in Table 1. The lake is an $HCO_3$–Ca–Mg type in the Shukalev classification, which is a typical karst water type [13]. The pH of the lake water is between 8.5 and 9.0, and $HCO_3^-$ is the main anion, with an average concentration of 188.4 mg/L, accounting for more than 90% of total anions [13]. The rocks exposed in the lake watershed are mainly limestone, dolomite, conglomerate, and basalt, and the main soil types are red earth, followed by brown earth. The vegetation belongs to southern subtropical evergreen coniferous broadleaved mixed forests.

**Table 1.** Results of the seasonal monitoring of hydrogeochemical indicators of Fuxian Lake in a hydrological year (1/2017~10/2017) [13].

| Sites | pH | T (°C) | DO (mg/L) | EC (μs/cm) | DO (mg/L) | [Ca$^{2+}$] (mg/L) | [HCO$_3$$^-$] (mg/L) | $p$CO$_2$ (Pa) | SI$_c$ | $\delta^{13}$C$_{DIC}$ (‰) |
|---|---|---|---|---|---|---|---|---|---|---|
| Surface water (No. = 12) | 8.5~9.0 [1] | 15.0~22.9 | 7.7~9.0 | 286.8~318.0 | 7.7~9.0 | 19.4~26.2 | 176.9~207.4 | 11.7~54.1 | 0.6~1.1 | $-1.2$~3.6 |
| | (8.8) [2] | (18.9) | (8.4) | (304.0) | (8.4) | (22.9) | (188.4) | (27.0) | (0.8) | (0.6) |
| | [2.2] [3] | [15.9] | [6.5] | [4.3] | [6.5] | [10.7] | [5.1] | [50.0] | [18.2] | [259.9] |

Note: [1] Minimum~maximum value; [2] Mean value; [3] Standard error.

The main primary producer of Fuxian Lake are phytoplankton [44]. The distribution area of submerged plants in Fuxian Lake is small, accounting for 2.4% of the water's surface area. The distribution depth is 0.5–14 m (average 4.3 m), and the distribution range is mainly along the lake margin and in the shallow water area (5.1 km$^2$). However, with respect to phytoplankton in Fuxian Lake, its species richness and diversity are relatively

low (mainly Chlorophyta followed by cyanobacteria, diatoms, dinoflagellates, and Charophyceae) due to the characteristics of freshwater oligotrophic lakes. It is noteworthy that the distribution range and total biomass of the macrophytes increased significantly in recent years. Moreover, the dominant species are *Potamogeton pectinatus*, *Vallisneria natans*, *Ceratophyllum demersum*, *Myriophyllum spicatum*, etc. [6,44]. However, Fuxian Lake (water quality: type I) has not been recognized as a eutrophic lake, the lake's ecosystem has not undergone significant evolution, and the tipping point of the steady state structure has not been broken [6,45]. These observations have also been reflected in the evidence of molecular depositions during the past 100 years [16,44].

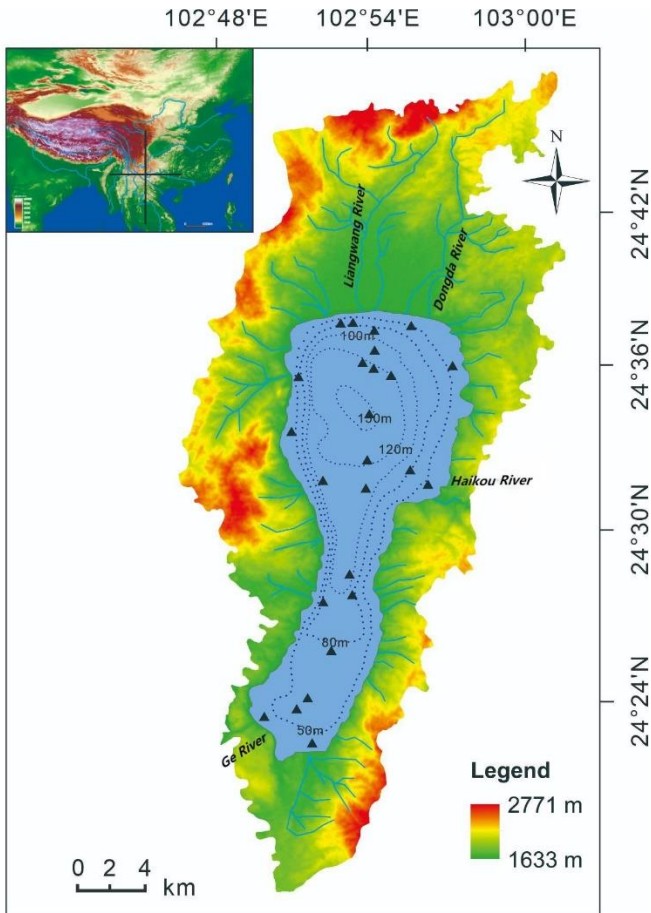

**Figure 2.** Location of Fuxian Lake and the sampling sites (black triangle) (data points are integrated with previous research [9,20,46,47]). The basin boundary and bathymetric contours (dotted lines) were from He et al. (2020) [15].

We compiled published data of the surface sediments on the TN and TOC contents, C/N (molar), and $\delta^{13}C_{org}$ values [9,20,46,47], which we used to conduct a spatial analysis using Kriging interpolation (Figure 3). It should be pointed out that, according to the sampling time recorded in the compilation documents, the age difference between the surface sediments was not significant. The age information of the sampling points was 2017 (number of data points: 3), 2015 (7), 2011 (13), and 2003 (2), respectively. The TN content of the surface sediments varied between 0.1% and 0.5% (average 0.3%), and their TOC content varied between 1.3% and 4.2% (on average 2.4%). There was a significant relation ($R^2 = 0.76$) between TN and TOC (Figure 4A), indicating that most nitrogen exists in sediments in the form of OM, and that the source of TN and TOC in the surface sediments of Fuxian Lake is consistent. Similarly, the spatial distribution of TN was close to that of TOC, and the content is lower than that in the adjacent eutrophic lakes (e.g., Dianchi Lake [33] and Xingyun Lake [48]) and typical lakes in other areas with strong human activities (e.g., Taihu

Lake [49]). The fact that the OM content of Fuxian Lake is lower than that of other lakes is mainly based on the small drainage area of Fuxian Lake and the relatively simple land-use mode. This area has not received serious disturbances from human activities. Fuxian Lake, as a large deep-water lake, is characterized by an oligotrophic state, low biodiversity, and relatively low water productivity, which together result in low OM contents. In addition, the results demonstrate the significant spatial heterogeneity of the TN and TOC contents of the surface sediments of Fuxian Lake, with higher values in the central–northern and southern parts of the basin and lower values in the central part (Figure 3). Evidence shows that the organic productivity near the major nutrient sources (e.g., inflow) is very strong, reflecting the riverine supply of nutrients [26,28]. The nutrient status, from the river's mouths to the central part of Fuxian Lake, is substantially affected by transport by the Dongda River and Ge River, which is manifested by an important decline in the sediment nutrient contents. However, when interpreting these patterns, the decomposition and preservation of organic matter should also be considered [16]. Fuxian Lake is a plateau deep-water lake (158 m), and deep water favors the preservation of organic matter, which has been verified by (1) the relatively low DO concentrations in water at the bottom [13,20], and (2) the fact that water depths are positively correlated with TN and TOC (Figure 4B).

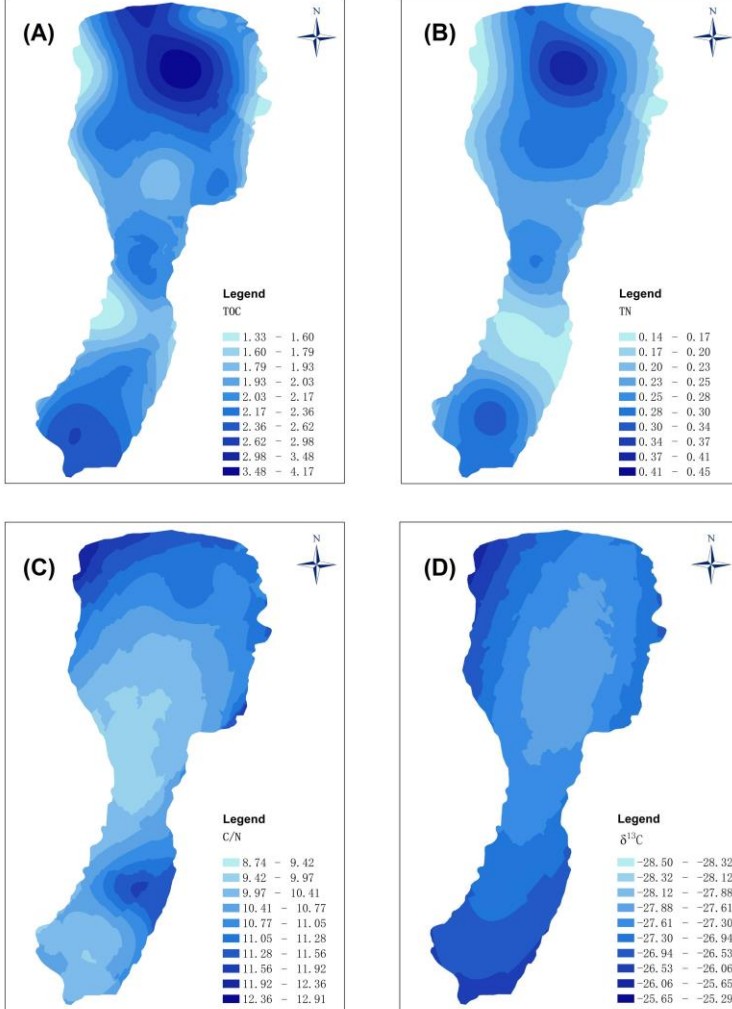

**Figure 3.** Spatial distribution of (**A**) TOC, (**B**) TN, (**C**) C/N molar ratio, and (**D**) $\delta^{13}C_{org}$ in the surface sediments of Fuxian Lake [9,20,46,47].

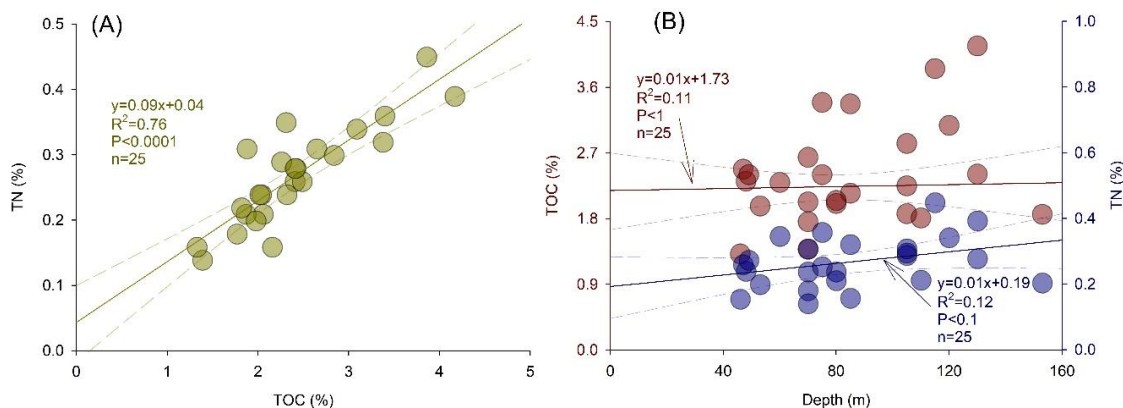

**Figure 4.** (**A**) Scatter plots showing the relationship between (**A**) TOC and TN and (**B**) water depth and TOC for the surface sediments of Fuxian Lake [9,20,46,47]. Best-fit linear regression lines were fitted to the relationships, and the $R^2$ values are shown. The curves represent the 95% confidence interval.

The $\delta^{13}C_{org}$ and C/N molar ratios of typical input endmembers (aquatic plants, terrestrial plants, and soil) of Fuxian Lake have been analyzed by many researchers, and the results are shown in Table 2. $\delta^{13}C_{org}$ displays an exponential decrease with water depth (Figure 5), ranging from −28.5 to −25.3‰, with a mean of −27.2% (Table 2). C/N ranged from 8.7 to 12.9, with a mean of 10.7 (Table 2), and there was no significant relationship with the water depth (Figure 5). Many researchers have suggested that $\delta^{13}C_{org}$ and C/N (molar) are significant indicators of changes in water depth [37,50,51]. Nevertheless, there are specific conditions for the application of any indicator, and these parameters cannot be used as paleo-hydrological indicators with general indicative significance, although they provided a significant contribution in past research studies. At the same time, it should be pointed out that the application of a reconstructed paleoenvironment still needs the joint support of a multi-index system established by other geochemical and paleo-ecological indicators [15,37]. In this study, the higher $\delta^{13}C_{org}$ values and C/N molar ratios in the shallow platform areas compared to the deeper offshore areas of Fuxian Lake can be attributed to the higher contribution of submerged macrophytes, while the contribution of phytoplankton mainly occurred in offshore sediments.

**Table 2.** The $\delta^{13}C_{org}$ and C/N for mixed plankton, submerged plant, terrestrial plants, soil, and surface sediments in Fuxian Lake.

| Input Endmember of Organic Matter | $\delta^{13}$C (V-PDB, ‰) | C/N | Reference |
| --- | --- | --- | --- |
| Mixed plankton | −23.6~−26.9 ($n$ = 7) | 6.91~9.05 ($n$ = 7) | [47] |
| Submerged plant | −13.7~−10.3 ($n$ = 7) | 11.1~12.8 ($n$ = 7) | [6] |
| Terrestrial plant | −13.0~−30.8 ($n$ = 8) | 16.0~89.0 ($n$ = 10) | [9,13] |
| Soil | −17.6~−27.5 ($n$ = 10) | 10.0~22.1 ($n$ = 10) | [9,13,20] |
| Surface sediment | −25.3~−28.5 ($n$ = 25) | 8.7~12.9 ($n$ = 25) | [9,13,20,46,47] |

The effectiveness of the $\delta^{13}C_{org}$ of surface sediments and the C/N molar ratios in elucidating their organic matter sources have been reported for lacustrine environments. According to a previous study in Fuxian Lake, the C/N molar ratio of the submerged plants ranged from 8.7 to 12.9. We could observe this phenomenon from Figure 6, that is, the C/N molar ratio of the sediment fell within the range of autochthonous OC, which was significantly different from the range indicated by allochthonous OC, reflecting the fact that lake sediments are mainly endogenous. A previous study found that the primary productivity of Fuxian Lake has been positively correlated with species richness since the 1950s. Moreover, the biomass compositions have indicated an increase in benthic species relative to phytoplankton [42]. $\delta^{13}C_{org}$ is often used to trace the source and contribution of carbon when the influence of decomposition mineralization or source overlap is not

considered [9,15]. The main controls on t $\delta^{13}C_{org}$ in most lake waters are as follows: (1) enhanced photosynthesis of aquatic organisms via isotopic fractionation mechanisms, which can introduce more negative $\delta^{13}C$ into the biological organism [52]; (2) the development of chemical autotrophic microbial communities intensified under eutrophication conditions, which introduced stronger fractionation effects than photosynthesis [53]; (3) inputs of terrestrial $C_3$ plants that introduce more negative values of $\delta^{13}C_{org}$, at an average of −27‰ [54]; and (4) enhanced degradation of organic matter, producing more $^{12}C$ then will be absorbed by phytoplankton, causing a less positive $\delta^{13}C_{org}$ in the organic matter [20].

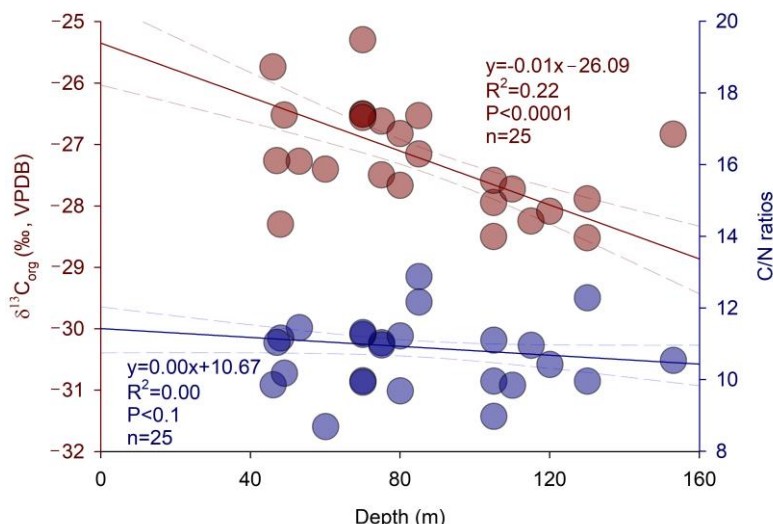

**Figure 5.** The $\delta^{13}C_{org}$ values and C/N molar ratios for the surface sediments from different depths in Fuxian Lake [9,20,46,47]. The curves represent the 95% confidence interval.

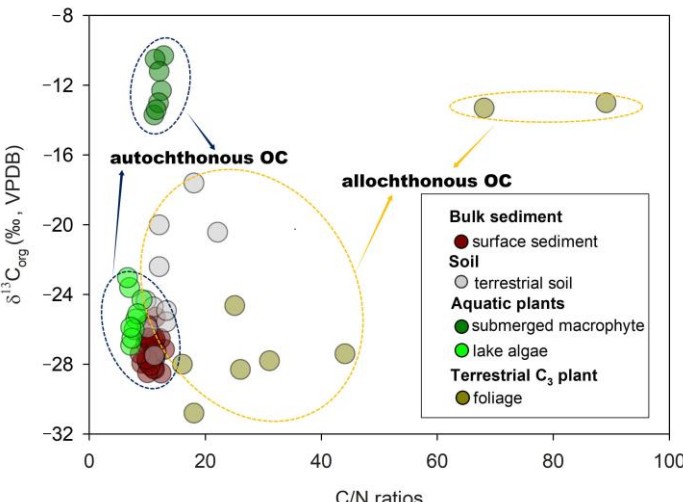

**Figure 6.** Scatter plot of the C/N molar ratio versus $\delta^{13}C_{org}$ for the surface sediments of various potential sedimentary organic matter sources for Fuxian Lake [9,20,46,47]. The sources include terrestrial plant materials, submerged plants, algae, and soil (the $\delta^{13}C_{org}$ values and C/N molar ratios for the sources were from Meyers (1994) [54]).

A comparison of C/N and $\delta^{13}C_{org}$ over a wide range of potential sediment sources of Fuxian Lake with the values of sediment samples makes it possible to identify the main sources of OM in the surface sediments [40]. Figure 6 shows a plot of $\delta^{13}C_{org}$ versus the C/N molar ratio for surface sediments or sources; it can be seen that the field for aquatic plants (autochthonous source) was different from those of modern terrestrial plants (allochthonous source). Based on the C/N molar ratios (Table 2), the low $\delta^{13}C_{org}$ values of

the surface sediments did not respond to mixing with terrestrial plant material (enriched in $^{12}$C). It is reasonable to conclude that the C/N molar ratios and $\delta^{13}C_{org}$ values reflect the main source of aquatic plants relative to the total organic matter content of the surface lake sediments. Similarly, other studies have confirmed that the OC in major lakes worldwide was mainly derived from aquatic plants (e.g., in Lake Taihu [49], Xingyun Lake [48], Bosten Lake [26], Dianchi Lake [33], and Ngoring Lake [15]). Accordingly, these studies shared the common recognition that autochthonous OC was the predominant contribution of the sedimentary organic carbon in Fuxian Lake, namely, the deposition of organic carbon by the biological carbon pump effect might be crucial to the carbon cycle in Fuxian Lake.

Research from Fuxian Lake suggests that organic matter is mainly derived from photosynthetic organisms, which strengthened the "DIC fertilization effect" and increased the autochthonous sources of OC. Although the quantitative differentiation between allochthonous OC and autochthonous OC has not been carried out yet, this is an attempt to examine the biological carbon pump effect in a water body. Our results indicate that at any timescale, aquatic photosynthesis coupled with rock weathering represents the current atmospheric $CO_2$ sink, with consequential climate impacts. The equation below reflects the nature of aquatic photosynthesis processes as well as the 'biological carbon pump' (BCP) concept in an aquatic ecosystem [14,17].

$$CaCO_3 + CO_2 + H_2O \rightarrow Ca^{2+} + 2HCO_3^{-} \overset{Photos}{\rightarrow} CaCO_3 + x(CO_2 + H_2) + (1-x)(CH_2O + O_2)$$

The BCP effect refers to the DIC absorbed by photosynthesis, namely, the 'DIC fertilization effect', and enhances carbon sequestration and aquatic biomasses [14,17]. Inorganic C promotes growth in submerged macrophyte biomass and primary production in karst lakes [1]. Therefore, an increased understanding of carbonate-weathering-related carbon sinks, with an emphasis on the mechanism of the carbon uptake by BCP effect and its influencing factors, is likely to provide valuable scientific support for attempts to regulate carbon cycling and climate.

## 5. Evaluation of the Nutrient Content of Surface Sediments

Numerous approaches have been used to evaluate the status and health of lake ecosystems. Considering the rapid increase in the accumulation rate of organic matter in the surface sediments of Fuxian Lake, an organic index and the organic nitrogen content of the sediments were used to evaluate the degree of organic contamination, which has been proven to be effective [42,55,56]. The organic index (=Org − C(%) × Org − N(%)) is commonly applied to assess the level of the organic contamination of lake surface sediments, and the organic nitrogen content (TN(%) × 0.95) is an important index for determining the degree to which nitrogen pollution affects the surface sediments of a lake [42,55,56].

The values of the organic index and the organic nitrogen content of the surface sediment samples are displayed in Table 3. The organic index of Fuxian Lake ranged from 0.19 to 1.65, with an average of 0.66. This can be classed as grade IV, which indicates that there is a potential risk of organic pollution. Greater values occurred in the southern marginal area, especially the northern area, as shown in Figure 2, which is close to the mouth of one of the major inflowing streams. The organic nitrogen index values were 0.14%~0.43%, with a mean of 0.25%. These results show that there is a serious risk of nitrogen pollution, up to grade IV, throughout the lake, and that the distribution of organic nitrogen content is consistent with that of the organic index (Figure 7). High values occurred in areas of the lake where the catchment is dominated by tourism, agriculture, and forestry, both of which can supply large quantities of allochthonous pollutants, and the deposition of various nutrients and biological residues may be an additional reason for the observed distribution of organic nitrogen pollution. Although modern monitoring shows that Fuxian Lake is an oligotrophic lake and that the contribution of organic matter (mainly, organic carbon) within the lake is mainly autochthonous, there is a potential risk of surface sediments becoming significantly contaminated by organic pollution, especially organic nitrogen pollution. Although this risk has yet to be fully realized, possibly because of the depth of

the lake, careful monitoring is required and control measures may need to be adopted in the future.

**Table 3.** Evaluation criteria for the types and grades of the organic index and organic nitrogen content of the surface sediments [42,55,56].

| | Organic Index | | | |
| --- | --- | --- | --- | --- |
| | **<0.05** | **0.05~0.20** | **0.20~0.50** | **>0.50** |
| Types and grades | Clean/I | Clean/II | Clean/III | Contaminated/IV |
| | **Organic Nitrogen Content** | | | |
| | **<0.0033** | **0.033~0.066** | **0.066~0.133** | **>0.133** |
| Types and grades | Clean/I | Clean/II | Clean/III | Contaminated/IV |

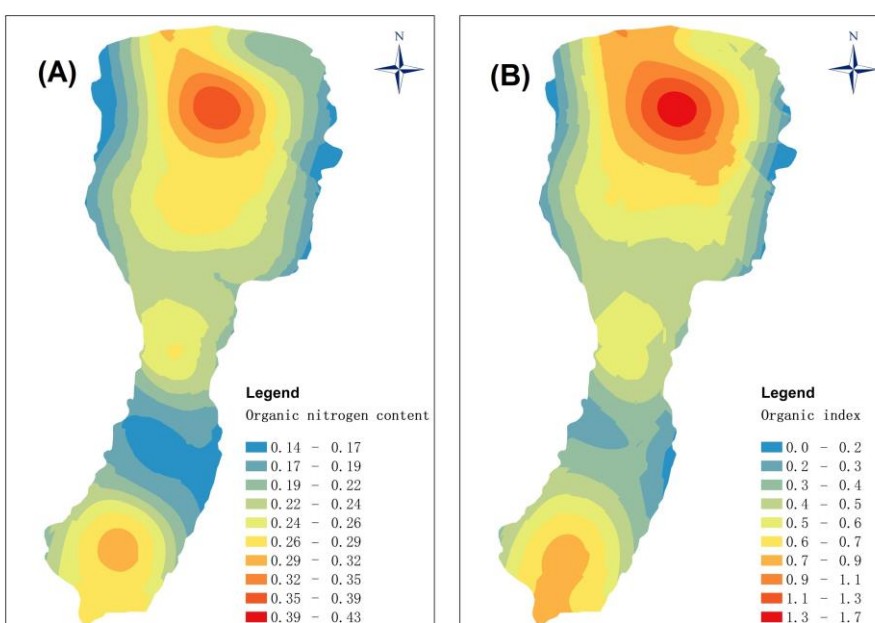

**Figure 7.** Spatial distribution of the organic nitrogen content (**A**) and organic index (**B**) for the surface sediment samples from Fuxian Lake [9,20,46,47].

## 6. Conclusions

We conducted a study of the spatial distribution of the source and composition of the organic components (carbon and nitrogen) of the surface sediments of Fuxian Lake on the Yunnan–Guizhou Plateau. Aquatic photosynthetic plants, especially phytoplankton, are the dominant organic matter source of surface sediments in Fuxian Lake. Decreases in the concentrations of TN, TOC, and C/N molar ratios occurred with increasing offshore distance, together with a decrease in $\delta^{13}C_{org}$, which was due to the greater contribution of aquatic macrophytes. Meanwhile, the autochthonous organic carbon in the surface sediments was characterized by lower $\delta^{13}C_{org}$ and C/N, suggesting that the biological carbon pump effect plays a significant key role in the stability of carbon sinks by coupling with carbonate weathering. The calculation of the organic index and the organic nitrogen content for these surface sediments indicates that there is a potential risk of organic pollution and organic nitrogen pollution, especially in the southern and northern nearshore areas. We suggest that careful monitoring of the degree of organic contamination of the surface sediments should be conducted, and it is possible that mitigation measures may be required in the future such as reducing the use of chemical fertilizers in cultivated land within the watershed, in order to improve service and management and maintain the water quality of Fuxian Lake.

**Author Contributions:** Conceptualization, Investigation, Writing, K.Z. and H.H.; Investigation, Data curation, D.L.; Resources, C.X. and X.H. All authors have read and agreed to the published version of the manuscript.

**Funding:** This study was financially supported by the National Natural Science Foundation of China (42007296); the Strategic Priority Research Program of the Chinese Academy of Science (XDB40020000); the Key Natural Science Research Project of Heifei Technology College (2021KJA11); and the China Postdoctoral Science Foundation (2021T140582).

**Data Availability Statement:** Not applicable.

**Acknowledgments:** We thank Jan Bloemendal for improving the English language.

**Conflicts of Interest:** The authors declare no conflict of interest.

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
