# Peer review of "Spatial Distribution and Sources of Organic Matter in the Surface Sediments of Fuxian Lake, SW China"

_water, doi:10.3390/w15040794_

Round 1
Reviewer 1 Report
Review of Spatial Distribution and Sources of Organic Matter in the Sur- 2 face Sediments of Fuxian Lake, SW China
This contribution is a review of works regarding organic matter sources and distribution at Fuxian Lake, China. However, it seems that authors conducted some experimental work as part of the review (see line 147). If these are new data, then a complete description of method employed should be added at least as supplementary material.
Keywords: Fuxian Lake. This Word is already in title. Please substitute for other keywords.
Line 281: change “organism” to “organisms”.
Reviewer 2 Report
In this paper author reviewed the data of organic matter distribution and sources in the surface sediments of Fuxian Lake, previous literature showed few studies that reported the organic matters and other minerals geochemistry. Most of them were reported by the authors of this manuscript. In this manuscript no new information had been reported that differs it from previously reported studied. This manuscript will be more attractive for readers if author compare same data with other lakes reported in China. Abstract is not very informative. Sentences were badly fabricated in whole manuscript.
Compare OM data of different lakes with Fuxian Lake. Also provide annotated bibliographical review of OM distribution of different lakes.
Provide temporal distribution of OM distribution in the surface sediments.
Rewrite introduction part from line 32-82, this rich text isn’t very informative, provide literature which explain background of the study. It will be more fruit full if you present graphically how biogeochemical cycle’s impacts surface sediments of lakes in one figure. Line 32 change “raise” to “rise”, line 34 change “freshwater systems” to “freshwater ecosystems”, give full form of “DIC”, “CCW” and so many mistakes.
Figure 2 was generated using studies ref. [9, 20, 46 and 47], all these studies published in different years it reflects temporal distribution rather than spatial.
Mention the sources of each figure in their captions.
Round 2
Reviewer 2 Report
Author's responses to the comments are not satisfactory, they must go through again and give satisfactory response to each comment.
Author Response
We thank the referee for their insightful comments which have greatly improved the manuscript. We have taken your comments on board and made every effort to address your concerns (including a reinterpretation and restructuring of the Figure Caption, introdution, and discussion).
"Please see the attachment."
Round 3
Reviewer 2 Report
Before acceptance please check the format, spell check, grammar and references.